## Perspective

community-based initiatives; education; global mental health; mental health

**Corresponding author:**
Adam D. Brown;
Email: brownad@newschool.edu

# Transforming mental healthcare in higher education through scalable mental health interventions

Adam D. Brown[1,2] , Nicole Ross[1], Manaswi Sangraula[1], Andy Laing[3] and Brandon A. Kohrt[4]

[1]Department of Psychology, New School for Social Research, New York, NY, USA; [2]Department of Psychiatry, New York University School of Medicine, New York, NY, USA; [3]Department of Social Work, University of The Bahamas-North, Freeport, Bahamas and [4]Division of Global Mental Health, Department of Psychiatry, George Washington University, Washington, DC, USA

## Abstract

A significant number of young people throughout the world are experiencing mental health concerns. Many young people will develop their first mental health concerns or will be managing their symptoms while enrolled in institutions of higher education. Although many colleges and universities are aware of the significant mental health needs among their students, the mental health and psychosocial needs of students often exceed the availability of resources and cultural and contextual barriers, such as stigma, may further impede access to care. Such gaps and barriers in mental health may lead to poor prognosis as well as negative educational and social outcomes. We propose that non-specialist delivered mental health and psychosocial interventions may play a critical role in reducing the gaps in care for students in higher education. In particular, non-specialist delivered care can complement existing specialized services to provide stepped models of care. Importantly, the adaptation and implementation of non-specialist delivered mental health and psychosocial support interventions in higher education may lead to innovative strategies for increasing access to care in this context, but may lead to adaptations that could apply to contexts outside of higher education as well.

## Impact statement

Higher education, such as college and university settings, has long symbolized opportunities for personal transformation, intellectual growth and learning, the discovery of new ideas, vocations, and the forging of long lasting personal and professional relationships. However, increased attention is being placed on the significant mental health challenges university students face. At the same time, there are often not enough mental health specialists within universities to address the numbers of students seeking help and stigma, previous negative experiences with counseling, and long wait-lists have been identified as additional barriers to care. How might universities begin to address these major gaps in mental healthcare? We propose that a vital strategy to increase capacity and reduce gaps in mental health support for university students is through the delivery of brief mental health interventions by nonmental health specialists. In particular, we recommend universities begin to contextualize and integrate nonspecialist delivered interventions that have been thus far employed primarily in the humanitarian context to the university setting. There is a growing evidence base for nonspecialist interventions such as Problem Management Plus and Self-Help Plus and which may enrich the availability of mental health resources for students. Universities may be ideally set up for the training of nonmental health specialists given the number of individuals who play supportive roles in student's lives. For example, student leaders, tutors, coaches, might be well positioned to integrate these strategies into their work. Although mental health specialists play a critical role in supporting the mental health needs students the burdens are far outpacing the availably of resources and the integration of evidence-based nonspecialist strategies can fill some of the urgent gaps in care in universities worldwide.

## Introduction

Higher education, such as college and university settings, has long symbolized opportunities for personal transformation, intellectual growth and learning, the discovery of new ideas, vocations, and the forging of long lasting personal and professional relationships. The role of higher

education institutions often extends well beyond learning and training by offering a wide range of services for students, including many that seek to support their health and wellbeing. Within these ecosystems, however, there is an urgent need for higher education to address the growing mental health needs of their students (Duffy et al., 2019). Students with mental health issues are often at greater risk for poor educational, social, health, and economic outcomes (Niederkrotenthaler et al., 2014; Scott et al., 2016). Moreover, for students from historically marginalized and underrepresented identities (LGBTQIA and BIPOC), colleges and universities can be a source of stress, due to factors such as separation from social support networks and a lack of culturally responsive services (Clark and Mitchell, 2018).

In recent years, there has been a growing recognition and the development of policies seeking to address mental health concerns among university students. For example, in the UK, national multi-sector guidelines have identified ways that barriers to mental health support are negatively linked to educational outcomes and, as such, have developed frameworks to reduce such challenges in accessing support (Universities UK, 2015).

Yet, recognition, policies, and frameworks continue to be met with a wide range of challenges that underscore the need to identify novel pathways to reach university students requiring mental health support. For instance, universities continue to respond to the mental health needs of students in light of the disruptions and negative consequences of COVID-19. Although the full mental health impacts of COVID-19 are yet to be known, initial studies indicate that the pandemic is associated with an increase in the onset of new mental health conditions as well as relapses or the worsening of existing conditions for current students (Chen and Lucock, 2022; Wood et al., 2022). For many students, COVID-19 was just one of many stressors and upheavals taking place in the distal and immediate landscapes of their lives. Throughout the world, young people are encountering complex and often unprecedented political, environmental, and social upheavals. In the face of forced migration and persecution, universities and colleges play an important role in providing intellectual havens for students – a role which is likely to increase due to protracted conflicts and climate-related threats (Casellas Connors et al., 2023).

Although the need to address mental health concerns in higher education is well documented, so too are the multiple factors that may impede care. Multiple factors such long wait times, overextended counselors, language barriers, peer- and self-stigma, cost, and the lack of representation from historically marginalized groups among mental healthcare providers, represent the challenges students face in terms of seeking care (Giamos et al., 2017; Broglia et al., 2021; Hingwe, 2021). Additionally, although there is considerable evidence that interventions such as mindfulness and cognitive behavioral therapy (CBT) are effective treatments for mental health concerns such as anxiety and depression in university students (Huang et al., 2018; Worsley et al., 2022), this population does not always seek care from specialists due to previous negative experiences in counseling, poor understanding of the existing services, and lack of availability for on campus counseling or long-term care (Bray and Born, 2004; Giamos et al., 2017).

## Nonspecialist delivered interventions in higher education

As universities continue to develop policies and invest in programs and resources for students, it is important to consider the role of nonspecialists in filling mental health gaps. One approach is to build on the growing body of work indicating the benefits of peer-led interventions, such as art, mindfulness, exercise, and general support. In fact, a review showed that a pooled analysis of non-CBT interventions, which included peer-based programs, were associated with a greater effect size in reducing depression and anxiety in university students when compared to mindfulness or CBT interventions (Huang et al., 2018). Furthermore, although not limited to higher education contexts, a recent scoping review examining the potential mental health benefits of peer-support (support provided by an individual(s) with a shared lived experience(s) for young adults, showed that peer-support was associated with greater levels of happiness, self-esteem, positive coping and lower levels of loneliness, depression, and anxiety (Richard et al., 2022)).

An important addition in addressing this urgent public health issue in higher education is to consider the potential role of brief, culturally and contextually adapted scalable, nonspecialist delivered forms of mental health and psychosocial support. Higher education would benefit from the growing knowledge base of nonspecialist delivered interventions that have thus far been primarily implemented in humanitarian contexts. Such scalable mental health interventions, either delivered or facilitated by nonspecialists, are often manual-based and are designed for broad uptake and dissemination – primarily focusing on strengthening existing coping skills to manage adversity and challenges. Current interventions range in format from individual (e.g., Problem Management Plus [PM+]), to group (Group PM+), to self-directed formats (Self-Help Plus [SH+]) (World Health Organization, 2017). Although longer term follow-up data are needed, there is now considerable evidence across a wide range of humanitarian contexts that nonspecialist delivered interventions are effective (van't Hof et al., 2020; Purgato et al., 2021).

Moreover, World Health Organization (WHO), strongly encourages the importance of cultural and contextual adaptation of the manualized interventions prior to implementation (World Health Organization, 2017). There already exist a number of frameworks such as DIME (Johns Hopkins Bloomberg School of Public Health, 2013) and the cultural adaptation and contextualization for implementation (mhCACI) (Sangraula et al., 2021) procedure that can be used to help in the tailoring of interventions to reflect the needs of the local university. This crucial practice may increase the likelihood of treatment-seeking behaviors, and the adoption of scalable interventions.

In many ways, higher education offers an ideal context for the integration of scalable interventions into an overall stepped model of care, in which given limited resources and barriers, there is a tailoring of resources based on the severity and intensity of mental health needs. Often, stepped care seeks to provide low-intensity interventions whereby the individuals are "stepped up" or "stepped down" based on the severity and acuity of their mental health concerns. Therefore, multi-level strategies ranging from health promotion, detection and referral training, the delivery of low-intensity interventions, and direct clinical care from specialists, are all integrated into the fabric of the community (Hermens et al., 2015; Cornish et al., 2017). While each institution has its particular structures and resources, many schools are comprised of students, staff, and faculty who routinely engage in the social and emotional lives of students, and who are therefore well positioned to promote mental health education and to implement mental health and psychosocial interventions. For example, student-led organizations, resident assistants (students who live with other students but assume a type of caregiving role), diversity and equity officers, as well as faculty who might work closely with student

mentoring and professional development, may be ideally positioned in their university to contribute to mental health promotion and literacy, and to the delivery of brief mental health and psychosocial interventions, along with making referrals to specialized services.

At The New School in New York City and at University of The Bahamas-North in Grand Bahama, we are currently disseminating PM+, a brief, nonspecialist delivered psychological intervention developed by WHO for adults with common mental health concerns and practical problems (Dawson et al., 2015) on several college and university campuses. Interviews with trainees and the use of the Ensuring Quality in Psychological Support (EQUIP, Kohrt et al., 2020) platform indicate that individuals with minimal training in mental health and psychosocial support exhibit core helping competencies central for delivering interventions such as PM+ (Pfeffer, 2023).

In pilot work with psychological first aid (PFA), an evidence-informed modular approach to help in the immediate aftermath of stressful and potentially traumatic events, we have found high levels of engagement in the training of staff and student leaders (Ross, 2023). Whereas PFA may be an important framework for responding to someone in immediate distress, other nonspecialist delivered interventions may be effective for university students experiencing ongoing distress or mental health concerns, such as anxiety and depression. In such cases, PM+ may be employed to strengthen coping skills to manage ongoing symptoms for a number of syndromes, as it is intended to be a transdiagnostic intervention (World Health Organization, 2017). Given that this intervention was meant to be delivered by nonspecialists, it could be well integrated into the work of existing roles (e.g., college life staff and resident assistants) and support structures (e.g., tutoring and professional development programs). Additional resources will strengthen the systems of student support already in place at many institutions. Critically, such strategies will reduce some of the burden on overtaxed health centers and serve as critical conduits for those with severe mental health issues and imminent risk to be connected to specialist providers. A growing number of studies also indicate that students from historically marginalized or underrepresented communities may be less likely to seek care at their university (Bouris and Hill, 2017; Sigal and Plunkett, 2023). Interventions delivered by nonspecialists who share similar lived experiences may help to increase engagement in mental healthcare.

## Future directions

Finally, integrating scalable interventions into mental healthcare in higher education will drive innovation. Mental health interventions are only as effective as students are willing to utilize them. The promotion of rigorous adaptation strategies within these highly creative and forward-thinking communities will likely generate a wide breadth of training and dissemination materials. In fact, new nonspecialist delivered mental health and psychosocial support interventions may be employed to provide a continuity of care in post-secondary contexts. Whereas higher education is a major life transition, so too are the challenges one might face in certain contexts after graduation. After the formal and informal supports associated with a university community, individuals may have even less access to care post-graduation. As such, new tools to facilitate the bridge between secondary and post-secondary contexts offers a novel area of work for

nonspecialist delivered interventions. Furthermore, the materials developed for students in higher education are also likely to have relevance for a wider range of contexts, and could serve as a catalyst for new strategies well beyond colleges and universities. Higher education can be pivotal positive experience in one's intellectual, professional, and personal trajectory. It is also evident that for many students, it is also a time in which they are in need of mental health and psychosocial support. Although a number of approaches may help to reduce gaps in mental healthcare in colleges and universities, the training of nonmental health specialists offers a novel evidence-based strategy for building much needed capacity for mental health services in higher education.

**Open peer review.** To view the open peer review materials for this article, please visit http://doi.org/10.1017/gmh.2023.29.

**Author contribution.** A.D.B., N.R., M.S., A.L., and B.A.K. contributed to the conceptualization, and writing and editing of the manuscript.

**Financial support.** This work was supported through a Fulbright Specialist Scholar grant awarded to A.D.B. and A.L. and a National Institutes of Mental Health grant, 1R01MH127767-01 awarded to B.K. and A.D.B.

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
