## [Reviewer Report]

January 9, 2023

Re: MHPSS in Higher Education Comment

Please find enclosed our submission to Global Mental Health entitled “Transforming Mental Healthcare in Higher Education Through Scalable Mental Health Interventions” as an Editorial.

It is now well established that mental health issues are an urgent public health issue in higher education. Rates of mental health issues were on the rise prior to COVID-19 and it appears that the pandemic has led to a worsening of mental health for many young people in college. Additionally, despite more attention and resources being allocated to support student mental health and wellbeing in higher education, significant barriers to care remain, with many students not being able to access care. In this Comment, we propose that brief scalable mental health intervention delivered by non-specialists can help to address some of the critical gaps in mental healthcare throughout higher education. Strategies such as “task sharing” have been found to be effective in humanitarian contexts, but have yet to be adapted and studied in a higher education settings. Such work would not only lead to innovative task sharing adaptations for young people, but such adaptations may then help to inform work beyond the university context.

This manuscript is original and has not been published elsewhere. The authors have no conflicts of interest to disclose with regard to the submitted work. All authors have agreed to the order of authors and to submission of the manuscript in its present form.

We very much look forward to hearing from you.

Sincerely,

Adam

Adam D. Brown

Vice Provost for Research

Associate Professor

Department of Psychology

New School for Social Research

---

## [Reviewer Report]

Reviewer Summary Statement:

This is an important perspective paper which highlights the critical importance of addressing the mental health needs of university students through integrating culturally sensitive, scalable interventions into mental healthcare in higher education. The authors develop their argument by noting some limitations in current services (e.g., barriers to accessing services) and propose alternative task-sharing strategies and the integration of existing programs as scalable approaches adaptable for higher education contexts. These are excellent points and appropriate for the perspective paper, however the arguments throughout would benefit from increased clarity and the addition of specific evidence as suggested in the comments below.

Abstract suggested revisions:

The sentence “*Additionally, the adaptation and implementation of task-sharing strategies in higher education settings may help to stimulate new research and thinking beyond colleges and universities as well.*” requires further clarification. For example, what is meant by task-sharing strategies as it is not mentioned or elaborated further in the main text of the paper.

Main text suggested revisions:

1. The organization of the paper would be greatly enhanced by using Cambridge Prisms: Global Mental Health journal’s perspective paper subheadings as recommended (e.g., introduction, main text divided into sections, conclusion and potentially future directions; see: https://www.cambridge.org/core/journals/global-mental-health/information/author-instructions/preparing-your-materials)

2. The focus and logic of the arguments in the paper could be improved with elaboration on some of the main points. Given the authors have additional space as per the guidelines (limit 2000-3000 words; currently 1700 words), areas to strengthen the manuscript have been suggested below.

- On page 3, the authors write *“Importantly, students indicate that they are pursuing information and support for mental healthcare online, but it remains difficult to assess the extent to how and if the rapidly growing marketplace of websites, podcasts, and social media platforms are contributing to positive mental health outcomes”*– Do the authors suggest that the programs should be offered online, that they can be provided by a non-mental health professional or both? It is suggested that the authors further develop this point.

- In addition, on page 4 the point is made *“Higher education would benefit from the growing knowledge base in off-campus contexts”*(reviewers’ underline emphasis), however, the authors should first include examples of non-specialist-delivered interventions for student mental health from within higher education contexts, especially as this is also a growing research area (e.g., Worsley et al., 2022).

- The following statement on page 4 would also benefit from elaboration regarding stepped models of care and how the proposed integration of programs fits with a stepped-care model. *“In many ways, higher education offers an ideal context for the integration of scalable interventions into an overall stepped model of care”*. Relevant citations should also be provided.

- There are sporadic mentions of differing levels of stress and distress (e.g., page 4: For more moderate levels of distress... or page 5: serve as critical conduits for those with severe mental health issues). It is unclear if the authors are suggesting the different programs (PM+, SH+) to address different issues based on severity. The argument needs clarification.

3. Furthermore, the paper could be significantly improved with the inclusion of citations supporting statements and/or effectiveness of the recommended alternative programs.

- For example, on page 3, the statement *“For many students, COVID-19 was just one of many stressors and upheavals taking place in the distal and immediate landscapes of their lives...”* appears to be missing a citation or the source of evidence for this claim.

- A central tenet of the perspective paper is that these alternative services have the potential to supplement existing services to the benefit of students. However, there is very little elaboration on what are common elements of these alternative services and the evidence for their effectiveness. Adding some detail around the outcomes associated with these programs and evidence of their effectiveness would be more persuasive. For example: when the authors note on page 4 *“Thus far, our preliminary results show that staff and students can be effectively trained, and that students receiving this find it beneficial in addressing their concerns.”* please specify, staff and students can be trained using what methods? Students receiving this program find it beneficial in what ways?

- Additionally, the authors highlight issues related to accessibility and lack of representation from historically marginalized groups, but do not provide evidence for how the suggested programs address these issues. It is suggested that the authors provide additional evidence on how these programs address these issues specifically, to make the link between this statement and the paper or remove the statement.

4. In the concluding comments on page 5, the authors end the paper with two imprecise statements, which weaken the focus of the argument. *“The promotion of rigorous adaptation strategies within these highly creative and forward-thinking communities will likely generate a wide breadth of training and dissemination materials. These materials can be used within the higher education context, but will also serve as a catalyst for new strategies well beyond colleges and universities.”*In place of these sentences, the authors are encouraged to emphasise the potential of integrating these scalable, non-mental health professional delivered services into post-secondary contexts in terms of access to support and wellbeing of a greater number of students, particularly marginalized students.

5. In summary, this perspective paper is timely, based on an excellent premise and suggests innovative solutions to address the pressing issue of student mental health and well-being in higher education contexts. With the suggested revisions to streamline and strengthen the arguments this paper has the potential to be a worthwhile contribution to Cambridge Prisms: Global Mental Health.

Editorial/Typographical :

“the mental health and psychosocial needs of among students often exceed”

“multiple factors such as long wait times,” - Please review manuscript for any typos/grammatical errors.

References should be reviewed for AMA (varying capitalization in article and journal titles)

---

## [Reviewer Report]

The broad concepts covered in this perspective piece are relevant and interesting, however the article needs substantial elaboration throughout. Several points made in this piece require more information with concrete examples and links to the literature. The abstract would also benefit from explicitly stating what the key messages are from this paper. The authors broadly list a few challenges and initiatives but not enough information has been provided to support the arguments being made despite there being plenty of relevant literature that could be cited. Examples include citing literature that supports the effectiveness of adapting interventions and the recent movement in the UK to bridge gaps between mental health services for university aged students.

Further information is also needed to explain the following interventions and strategies mentioned in the paper:

- Cultural and historically marginalised students - please elaborate and provide concrete examples about this group, their unique challenges and interventions

- Stigma is a broad statement and could include many types with differ t impacts. More detail about student challenges is needed throughout.

- Adapting and implementing interventions - these concepts are vague and justifying their use would be strengthened by providing concrete examples and links to the literature

- Task-sharing - elaborate or define what this is and provide examples.

- Non-mental health specialists - this includes a long list of people and some of whom would be inappropriate to work with students in distress. Please define, elaborate and provide concrete examples.

- Unwelcoming campus climates - please define and elaborate

- Some statements require citations - please check throughout

- Non-specialist forms of support - this is too vague and needs concrete examples

- PM+ intervention - some elaboration on what this is would help support the authors argument. In the current format, readers would have to read the original WHO paper to gauge what the interventions involves and how it’s been implemented in the named institution. This is particularly important given the results reported in this piece. The implications of the results cannot be gathered without this further information.

While there are several required edits to this piece I do believe it will shape up to be an interesting perspectives piece. More clarity and information throughout the article will adequately support the arguments being made.

---

## [Reviewer Report]

Dear Drs. Bass and Shidhaye,

My co-authors and I are sincerely grateful for overseeing this manuscript. We are thankful for the careful reviews and input provided by the reviewers. We have responded to each of the points raised in the review process and believe the manuscript is much stronger as a result.

Thank you very much again.

Best,

Adam

---

## [Reviewer Report]

We thank the authors for their careful consideration of reviewer feedback and associated edits which have strengthened the clarity of arguments within the manuscript. The addition of further research, elaborations regarding the WHO programs, as well as the re-organization using subheadings contributed to a substantially improved manuscript. However, we are requesting some final minor revisions, most of these revisions focus on the new material that has been added. We have listed these below including the page numbers.

General Feedback:

A thorough editorial review is needed to correct typos, grammar, punctuation, and capitalization throughout the manuscript.

Page 1 Abstract: “*the mental health and psychosocial NEEDS OF AMONG students often exceed the availability of resources and cultural and contextual barriers, such as stigma, may further impede access to care.*”

Page 4: “IntroUDCtion”

Page 4: “*The role of higher education institutions often extends well beyond learning and training by offering a wide range of services for students, including many that seek to support THE health and wellbeing.*”

Introduction:

Page 6: We recommend not using the term ‘recent’ (e.g., a recent review) when the review is from 2004

Page 4: We would suggest the authors consider using a more inclusive acronym than “LGBT”, alternatively using a broader, more current acronym such as “LGBTQIA+”

Page 4: Please clarify the sentence below. Specifically, do the authors mean that stressors NEGATIVELY associated with help seeking are linked to ADVERSE educational outcomes?

“*For example, in the UK, national multi-sector guidelines have identified ways that stressors associated with help seeking are linked to educational outcomes and have developed frameworks to reduce barriers to care (Universities, UK., 2015).*”

Non-Specialist Delivered Interventions in Higher Education:

Page 6: The following statement requires clarification/elaboration. Were the peer-based programs comparable to the CBT or mindfulness programs? What was the content of the programs? If the content is comparable, it would be important to highlight that this is evidence that non-specialist delivered programs can be more effective than traditional programs. “*In fact, a recent review showed that peer-based programs were associated with a greater effect size in reducing depression and anxiety in university students when compared to mindfulness or cognitive behavioral therapy (CBT) interventions (Huang et al., 2018).*”

Contextualizing and Integrating into Higher Education Systems:

Page 9: We recommend the authors not use acronyms that may be unfamiliar to the audience (e.g., EQUIP). “*Interviews with trainees and the use of the EQUIP platform indicate that individuals with minimal training in mental health and psychosocial support exhibit core helping competencies central for delivering interventions such as PM+ (Pfeffer et al., 2023).*”

Future Directions:

Page 10: We feel that the authors could strengthen the final concluding statement to focus more directly on the main thesis of the manuscript, i.e., the potential change in university service provision rather than the extension beyond colleges and universities. "*Furthermore, the materials developed for students in higher education are also likely to have relevance for a wider range of contexts, and could serve as a catalyst for new strategies well beyond colleges and universities*.”

Reference:

Page 11-14: Correct APA formatting in the reference list (italics, hanging paragraph, doi, punctuation)

---

## [Reviewer Report]

Dear Editors,

Thank you very much for this encouraging news. Please see our responses to the reviewer below. We hope that you feel that we have now addressed their remaining suggestions. Thank you again.

Reviewer: 1

Page 1 Abstract: “the mental health and psychosocial NEEDS OF AMONG students often exceed the availability of resources and cultural and contextual barriers, such as stigma, may further impede access to care.”

This has been corrected.

Page 4: “IntroUDCtion”

This has been corrected.

Page 4: “The role of higher education institutions often extends well beyond learning and training by offering a wide range of services for students, including many that seek to support THE health and wellbeing.”

This has been corrected.

Introduction:

Page 6: We recommend not using the term ‘recent’ (e.g., a recent review) when the review is from 2004

This has been corrected.

Page 4: We would suggest the authors consider using a more inclusive acronym than “LGBT”, alternatively using a broader, more current acronym such as “LGBTQIA+”

This has been corrected.

Page 4: Please clarify the sentence below. Specifically, do the authors mean that stressors NEGATIVELY associated with help seeking are linked to ADVERSE educational outcomes?

“For example, in the UK, national multi-sector guidelines have identified ways that stressors associated with help seeking are linked to educational outcomes and have developed frameworks to reduce barriers to care (Universities, UK., 2015).”

This has been clarified.

Non-Specialist Delivered Interventions in Higher Education:

Page 6: The following statement requires clarification/elaboration. Were the peer-based programs comparable to the CBT or mindfulness programs? What was the content of the programs? If the content is comparable, it would be important to highlight that this is evidence that non-specialist delivered programs can be more effective than traditional programs. “In fact, a recent review showed that peer-based programs were associated with a greater effect size in reducing depression and anxiety in university students when compared to mindfulness or cognitive behavioral therapy (CBT) interventions (Huang et al., 2018).”

We now more clearly state that the comparison was part of a larger pooled analysis of non-CBT interventions. We also included an additional study showing the benefits of peer support on young adults, which includes college students. However, the pooled analysis can not be compared specifically to a CBT intervention given the range of interventions included in this category. However, we hope this will address your careful comment. Thank you.

Contextualizing and Integrating into Higher Education Systems:

Page 9: We recommend the authors not use acronyms that may be unfamiliar to the audience (e.g., EQUIP). “Interviews with trainees and the use of the EQUIP platform indicate that individuals with minimal training in mental health and psychosocial support exhibit core helping competencies central for delivering interventions such as PM+ (Pfeffer et al., 2023).”

This has been corrected.

Future Directions:

Page 10: We feel that the authors could strengthen the final concluding statement to focus more directly on the main thesis of the manuscript, i.e., the potential change in university service provision rather than the extension beyond colleges and universities. "Furthermore, the materials developed for students in higher education are also likely to have relevance for a wider range of contexts, and could serve as a catalyst for new strategies well beyond colleges and universities.”

We added a new final sentence to bring this back to the main thesis of the paper.

Reference:

Page 11-14: Correct APA formatting in the reference list (italics, hanging paragraph, doi, punctuation)

We believe the formatting is now aligned with the guidelines.

---

## [Reviewer Report]

We are pleased to inform the authors that following the implementation of the requested edits, we are recommending the manuscript for publication. The manuscript provides valuable insights and offers significant contributions as a perspective paper. It successfully highlights the critical importance of addressing the mental health needs of university students and underscores the urgency of integrating culturally sensitive, scalable non-specialist delivered care within higher education settings.